

# Multiple execution of the same MPI application to exploit parallelism at hotspots with minimal code changes: a case study with FESOM2-Iceberg and FESOM2-REcoM

Kai Himstedt[1]

[1]German Climate Computing Center (DKRZ), Hamburg, Germany

*Correspondence to*: Kai Himstedt (himstedt@dkrz.de)

**Abstract.** For a typical climate model, parallelization based on a domain decomposition is a predominant technique to speed up its computation as an MPI (Message Passing Interface) application on an HPC (High Performance Computing) system. In this contribution, it is shown how the potential of simultaneously executing multiple instances of such an MPI application can

be exploited to achieve a further speedup with an additional parallelization of suitable compute-intensive loops. In contrast to a parallelization based on OpenMP (Open Multi-Processing), no special synchronization effort is required if MPI calls occur in the iterations of the original loop. Splitting the work at such hotspots between the instances represents an independent level of parallelization on top of the domain decomposition. The simple implementation can be performed within the familiar MPI world, where the climate model can largely be considered as a black box. Outside of the hotspots, however, the same

computations are performed in all instances. Some examples will show that such a conscious acceptance of redundant computations for parallelization approaches is quite common in other disciplines to reduce the time-to-solution. These approaches thus also represent the main inspiration for the approach presented in this contribution. Experimental results show for the example of the additional parallelization of an iceberg and a biogeochemical model, each embedded into FESOM2, how the time-to-solution can be further reduced with a small number of instances at appropriate efficiency. With the non-

parallelized part outside of hotspots, however, the meaningful utilization of a larger number of instances will not be easily possible in practice, which will be explained in more detail in some efficiency considerations with the reference to Amdahl's Law. Nevertheless, the implementation of the approach for other simulation models with similar properties seems promising, if the further reduction of the time-to-solution is in the focus, but a limit for the scalability based on the domain decomposition is reached.

## 1 Introduction

In engineering and natural sciences, there is an ongoing demand to speed up simulation models in order to bring the simulation results into an increasingly better match with reality. Many simulation models are based on the definition of boxes or meshes to define geometric regions. Domain decomposition, which breaks down the entire region into smaller parts, is a predominant technique to enable their parallel computation on HPC systems. Typically, the parallelized simulation model is executed as a



job on an HPC cluster system with the help of a workload manger like SLURM (Simple Linux Utility for Resource Management) (Jette et al., 2003; SchedMD, 2022). Depending on the number of cluster nodes requested, the sub-domains are assigned to the processors or processes, respectively, available with the job. This is especially true for a climate model like FESOM2 (Finite-volumE Sea ice-Ocean Model) (Danilov et al., 2017), which in this paper is the basis of the test environment.

For the numerical computations, it is usually necessary to store at the boundaries of the sub-domains also some information
about the boundaries of their neighboring sub-domains in so-called halo regions. With the halo exchange at the synchronization points, the data from neighboring remote processes are made available in the halo region of a local process. For the implementation of such data exchanges, MPI is the de-facto standard. Associated with the halo exchange between MPI processes is an overhead for synchronization (e.g. waiting for partial results) and communication (e.g. sending and receiving partial results). If the communicating MPI processes are located on the same node, this can be mitigated somewhat, since
access to shared memory with its high bandwidth and low latencies can then be used for the message passing.

The terms core and processor – in its classical meaning of a physical unit capable of executing a program – are used synonymously in this paper. With nowadays common multi-core CPU architectures, a considerable number of processors with the potential of shared memory access is available on a single cluster node. Hybrid approaches provide a way to exploit such resources at the intra-node level more effectively than just for accelerating MPI communication. Using multiple threads per
MPI process to parallelize compute-intensive loops within the same sub-domain is a typical example. In the past, the number of cores per node has increased steadily, as is reflected in the evolution of the list (established in 1993) of the 500 most powerful supercomputers, which is updated twice a year (TOP500, 2023). While an increase in computing power by increasing the processor clock rates has not taken place for some time, or at least barely, it can be assumed that this multi-core trend will continue in the future.

In this paper a different approach is presented that provides an independent level of parallelism based on pure MPI communication and for which an existing simulation model can largely be considered as a black box. The basic idea is to simply start an instance of the same simulation application, which has already been parallelized based on a domain decomposition, multiple times. Each of these instances contains exactly the same set of MPI processes (ranks), which will also be referred to as a group in the following. The work at hotspots, in particular for a compute-intensive loop, can now be easily
divided between the same group members (ranks) of different instances, provided that the iterations of the loop can be executed concurrently without synchronization. It should be briefly pointed out here that in particular already existing MPI calls are supported within such a loop without any additional (synchronization) effort being required for this, in contrast to OpenMP. After all partial results have been merged at the end of the loop in such a way that all instances have the same state of internal data again, the independent execution of the instances continues until the sequence repeats when another hotspot, parallelized
in this way, or the same one, e.g. in the next simulation step, is reached. Outside of such hotspots, redundant computations are consciously accepted as a compromise in order to achieve a speedup.

Section 2 discusses how similar ideas have been used in the past in other disciplines to achieve speedups for parallel applications based on the acceptance of redundant computations. In Sect. 3 the new approach is presented in detail. The



implementation was carried out in the context of the PalMod project (PalMod, 2022), and therefore, two experiments were
performed taking an iceberg model (Rackow et al., 2017; Ackermann et al., 2022) and the biogeochemical model REcoM
(Regulated Ecosystem Model) (Gurses et al., 2021) – in each case embedded into FESOM2 – as an example. In the first series
of experiments, the suitability of the new approach for parallelizing the main loop over all icebergs (executed in each simulation
step in each sub-domain) was examined. Further series of experiments were performed to determine how the expensive
computations for the tracer transport can be parallelized when REcoM is used in FESOM2 – also in comparison to OpenMP.
Section 4 contains the results of the experiments. Finally, the major insights are concluded in Sect. 5, where some future work
is also pointed out.

## 2 Accepting redundant computations in parallel applications

It seems quite natural to accept redundancies to improve the fault tolerance of a system, e.g. by combining multiple physical
storage components to a Redundant Array of Inexpensive Disks (RAID) system (Patterson et al., 1988) or by using techniques
that can be applied to enable an automated MPI application recovery (Bouteiller, 2015). There may also be performance
improvements associated with it. RAID systems, for example – quasi as a side effect – can increase data throughput because
it is possible to read from several disks simultaneously. An improved fault tolerance for MPI applications based on redundancy
can in turn reduce the effort required for the restart handling, which can potentially reduce the time-to-solution. For primarily
achieving further speedups with parallel systems, on the other hand, conscious acceptance of redundant computations is not
very common, especially not – to the best of our knowledge – in the context of HPC applications for climate modeling, which
are typically based on a domain decomposition. The main reason for this is assumed to be that applications in the past have
still shown an acceptable strong scaling behavior on typical HPC systems with domain decomposition. The limit of scalability
for steadily increasing numbers of processors had not yet been reached (see also Amdahl's Law, e.g., in Gustafson, 2011) and
further approaches for speeding up the models have therefore not been in focus.
Nevertheless, in some disciplines it is also common to accept redundancies or overheads consciously in order to achieve a
further speedup at all and thus in particular to reduce the time-to-solution. This is especially true for the scientific field of
distributed (game-)tree search, where algorithms like the alpha-beta search (Knuth and Moore, 1975) have a so-called
sequential nature and have been notoriously difficult to parallelize (see also Marsland and Gao, 1995 p. 2). In the following,
we present some classical approaches, mainly related to distributed tree search, which focus on the multiple parallel execution
of the same functionality to achieve a speedup at the cost of redundant computations, and thus have provided the main
inspiration for the approach presented in this paper. With an increasing number of available cores to reduce the time-to-
solution, the new approach can be used as a further level of parallelization, especially if the regions of a climate model can no
longer be meaningfully broken down into smaller parts by means of the existing domain decomposition.
    For a problem from the field of number theory related to integer factorization of (large) natural numbers Watt (1986)
describes his Parallel Polyalgorithm as follows: A number of well-established algorithms are available for integer factorization,





but their runtimes can differ greatly depending on the given number. This circumstance can be exploited by running the algorithms simultaneously, so that the result is available as soon as the algorithm most suitable for the given number has found the solution. Osborne (1990), in the context of his classification of such approaches, calls this variant Multiple-Approach. Segre et al. (2002) describe Nagging as an approach for parallelizing various typical artificial intelligence (AI) search methods,

which is based on the Parallel-Polyalgorithm/Multiple-Approach idea. A nagger (as employee) and its employer examine competitively an identical part of the search space, but in doing so they can proceed in different ways, which also leads to the expectation of different solution times. For A* search, a speedup of 1.51 with 2 processors and 4.95 with 64 processors is reported.

Weill (1995, 1996) has implemented a parallel game-tree search method, which can be described in a somewhat simplified

way as follows: On all processors, the same analysis for the root position of the search tree is initially started simultaneously. The goal is to speed up the search by allowing all processors to benefit from the constant interplay filling concurrently a shared hash table with search results of subtrees and accessing already generated hash table entries. The shared hash table serves as a kind of a cache, so that a subtree does not need to be searched again if its result is already available in the hash table. In his experiments with chess, he observes a speedup of about 15.5 with 32 processors.

As it was briefly mentioned at the beginning of this section, approaches where redundancy is primarily used to improve the stability of MPI applications can also be viewed from the perspective of an associated performance improvement. Elliot et al. (2012) evaluated the combinations of two techniques for improving fault tolerance in the context of the steadily increasing number of cores in HPC cluster systems. A traditional and typical technique to mitigate the problem of faults for long-running HPC applications is to write checkpoints periodically and, in the event of a fault, to restart and redo the lost work after the last

checkpoint. In this context, they refer to studies that have shown that applications running at a large scale spend more than 50 % of their total time on this. With redundancy as another technique, where multiple processes perform the same computations, the error rate per se can be reduced because if one process fails, a replica (as it is called in their contribution) can take over the execution. For their studies, they combined checkpoint/restart handling with redundancy at different degrees to minimize the total wall clock time and resource utilization of HPC applications. For an example with about 80,000 processes,

they state that although redundancy requires twice the number of processing resources, two jobs with a wall clock time of 128 hours can be completed in the time it would take to complete only one such job without the use of redundancy. They point this out as a special advantage for the upcoming exascale HPC systems, where the job throughput in the sense of capability computing is in the focus.

Losada et al. (2020) in their contribution on fault tolerance of MPI applications in exascale systems discuss the promising

approach of the User Level Failure Mitigation (ULFM) interface from the MPI forum. ULFM enables the implementation of robust MPI applications that can detect and react to failures without aborting the application. Unlike many other MPI extensions where fault tolerance is implemented close to the MPI layer and largely hidden from the application (e.g. rMPI (redundant MPI)) (Ferreira et al., 2009), the ULFM interface for fault detection and reconfiguration of communication structures in the event of a fault is located at the application layer. Redundancies are accepted in ULFM, for example, when





processes are kept available as hot spares that can take over the work of failed processes. In their paper, Losada et al. provide a broad overview of a wide variety of contributions for which the use of ULFM is the basis for fault tolerance. For Bland et al. (2014) they highlight that using ULFM compared to a traditional checkpoint/restart handling for an iterative refinement stencil code using the Monte Carlo Communication Kernel can improve the total time to completion by as much as 75 %.

## 3 Multiple execution of the same MPI application

Executing the same MPI-parallelized application multiple times in groups (hereafter also referred to as MULTIEXECMPI), with conscious acceptance of redundant computations to achieve a speedup, is quite similar in simplicity to the approach of Weill (1995, 1996) in the previous section. While Weill achieves a speedup with parallel filling of a hash table, MULTIEXECMPI at suitable hotspots enables the easy splitting of work between the groups to achieve a speedup. The reason why the splitting of the work is so particularly easy is that the multiple execution of groups $G_1, ..., G_m$ at the moment of reaching a hotspot ensures

that the contents of all application-related data structures of MPI processes $P_i$ with $i = 1, ..., n$ in group $G_1$ are identical to the corresponding contents of all MPI processes $P_i$ with the same $i$ in each of the groups $G_2, ..., G_m$. This may seem similar to the MPI extensions presented at the end of the previous section, where replicas (see also Elliot et al., 2012) are kept in reserve to improve fault tolerance. With these extensions, however, a speedup is achieved rather indirectly and usually requires the event of a fault so that a performance advantage can be achieved at all compared to not-using redundancy. With MULTIEXECMPI,

in contrast, the direct achievement of speedups is in the foreground. If required, it will be straightforward to combine MULTIEXECMPI with an approach to improve fault tolerance, especially if it is nearly transparent to the MPI application.

A suitable alternative for the basic idea of MULTIEXECMPI is the parallelization of a compute-intensive loop based on OpenMP. For the iterations of the loop, it is assumed that they can be executed concurrently without synchronization, i.e., there must be no data dependencies between the statements of different iterations (see also explanations for satisfying the

Bernstein conditions, e.g. in Feautrier, 2011). While with OpenMP shared memory access is available for the threads to store the partial results in common data structures, with MULTIEXECMPI corresponding MPI calls are required to merge the partial results across process boundaries for all MPI processes $P_i$ with the same $i$ in each group. After the merge (e.g. via MPI_Bcast), the MPI processes of different groups can continue their computations independently of each other until the next hotspot is reached. Extensions of MPI for a shared memory access like MPI SHM (MPI Shared Memory) (see also Brinskly et al., 2015)

are not within the scope of this paper.

A loop can also contain phases with MPI communication, so that MPI calls can occur concurrently in the iterations. An advantageous feature of parallelizing such a loop on the basis of MULTIEXECMPI is that no special synchronization effort is required for this. The MPI processes of a group can potentially communicate with each other arbitrarily without this having an influence on the communication of MPI processes in other groups. (For the sake of simplicity, a possible minor effect on

the communication bandwidth is neglected here.) For parallelizing such a loop with OpenMP, MPI environments usually provide a thread support for the MPI communication. However, for the degree of support and the performance of the related



implementation there is a strong dependency in practice on the MPI environment actually used. The thread support level is requested with the MPI_Init_thread call of a hybrid application and determines if, for example, only one thread at a time may make MPI calls (level MPI_THREAD_SERIALIZED), or if multiple threads may make MPI calls concurrently at any time (level

MPI_THREAD_MULTIPLE). In the latter case of implicit synchronization, typically quite a few internal MPI data structures (e.g. the request message queue) are locked, to which the various threads need common access to handle potentially arbitrary MPI communication. Compared to an explicit synchronization, which can be done with knowledge about the actual requirements with respect to the communication structures of the MPI application in a correspondingly fine-granular manner, the performance of implicit synchronization may well be lower in our experience. This disadvantageous effect when using

MPI_THREAD_MULTIPLE was about the same in OpenMPI and Intel MPI environments. Durnov and Oertel (2019), in the context of recent features of the Intel MPI environment, discuss a programming model (named MPI_THREAD_SPLIT) that can be used to improve the performance of concurrent MPI calls in the context of implicit synchronization. For this programming model, the MPI application must satisfy certain conditions, such as: Cross-thread access to MPI objects is not allowed and any request created in a thread must not be accessed by other threads. The conditions will often be easy to satisfy in practice, but

this programming model is currently only supported for Intel MPI environments.

The comparison between the parallelization of a loop with OpenMP and MPI shall not be discussed in further detail here, because an essential aspect with MULTIEXECMPI is the avoidance of the additional complexity of a hybrid approach. This additional complexity arises essentially for the private/shared differentiation of data structures, for handling concurrent MPI calls, for setting up the OpenMP build and runtime environment, for debugging a hybrid application, for acquiring OpenMP

know-how, etc. Using the example of FESOM2 with the extension of the biogeochemical model REcoM (FESOM2-REcoM for short), below a more detailed description is given of how MULTIEXECMPI allows an existing simulation model to be largely viewed as a black box and supports the parallelization of a compute-intensive loop as a hotspot with minimal code changes. Figure 1 shows for the example of FESOM2-REcoM with $m$ groups and $n$ MPI processes per group how the splitting of the $m \times n$ MPI processes in the startup phase is achieved by simple MPI communicator splits.



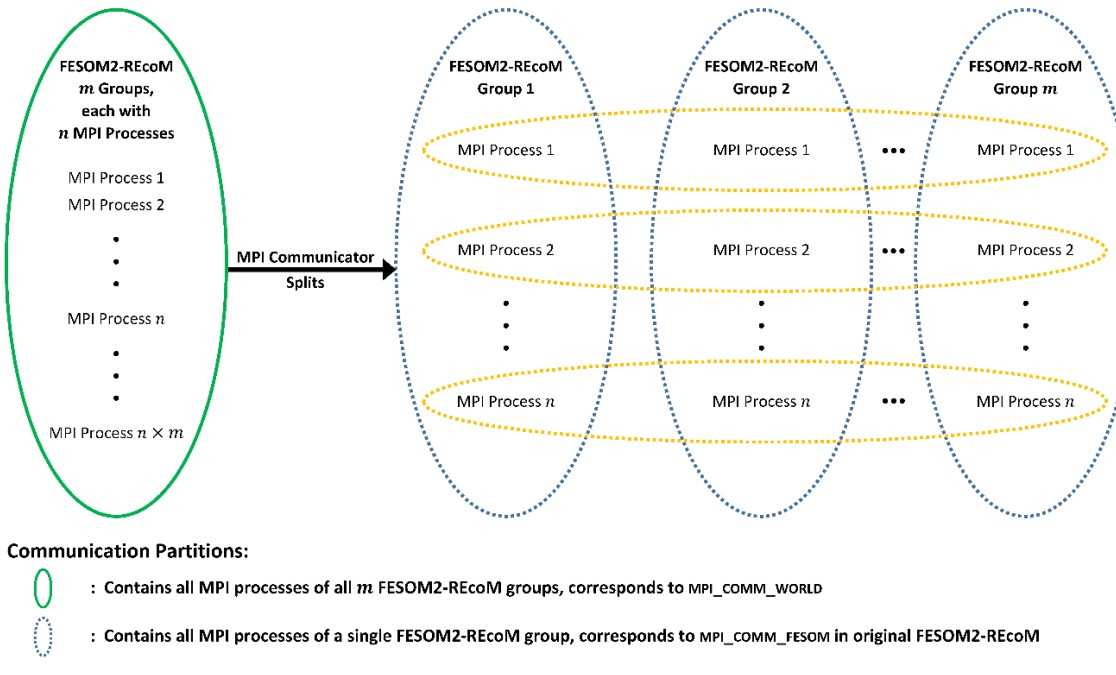


**Figure 1.** Multiple execution of an MPI application during startup phase.

For some of the experiments, which will be presented in more detail in Sect. 4, FESOM2-REcoM was additionally coupled to the ECHAM6 atmosphere model (Stevens et al., 2013) via OASIS (Ocean Atmosphere Sea Ice Soil) (Craig et al., 2017). For the startup phase, the coupling means nearly no additional complexity for the implementation. Instead of the

MPI_COMM_WORLD communicator, which now also contains the additional MPI processes of ECHAM6, FESOM2-REcoM simply uses the communicator as the starting point of the splitting, which as a result of the initialization of the coupling contains the $m \times n$ MPI processes available for FESOM2-REcoM. In the FESOM2-REcoM sources themselves almost no implementation effort was required to handle the coupled case. For the extension of the OASIS coupler, which only required a few lines of code, just an optional parameter for the number of groups had to be additionally handled in the initialization call.

For applications which do not specify this optional parameter during initialization of the coupling, such as ECHAM6 in the present case, the extension of the OASIS coupler is completely transparent (i.e. a rebuild of ECHAM6 is required, but no source code change).

In the startup phase, no distinctions are made between the groups for reading the different input files (e.g. mesh description, hydrographic data, ...) with which the initial conditions of the model are described (i.e. all groups read all files). The same

applies for the reading of restart files if the simulation has to be resumed at the last checkpoint after an event of a fault. Since these are one-time read operations and the number of groups will be rather small, no special performance disadvantages are to be expected. When writing result or restart files, on the other hand, only group 1 takes over these tasks for reasons of simplicity.



Figure 2 shows for the FESOM2-REcoM example how the tracer transport computations, which make up about 80 % of the total computing time of a simulation run, can be accelerated with 3 groups.

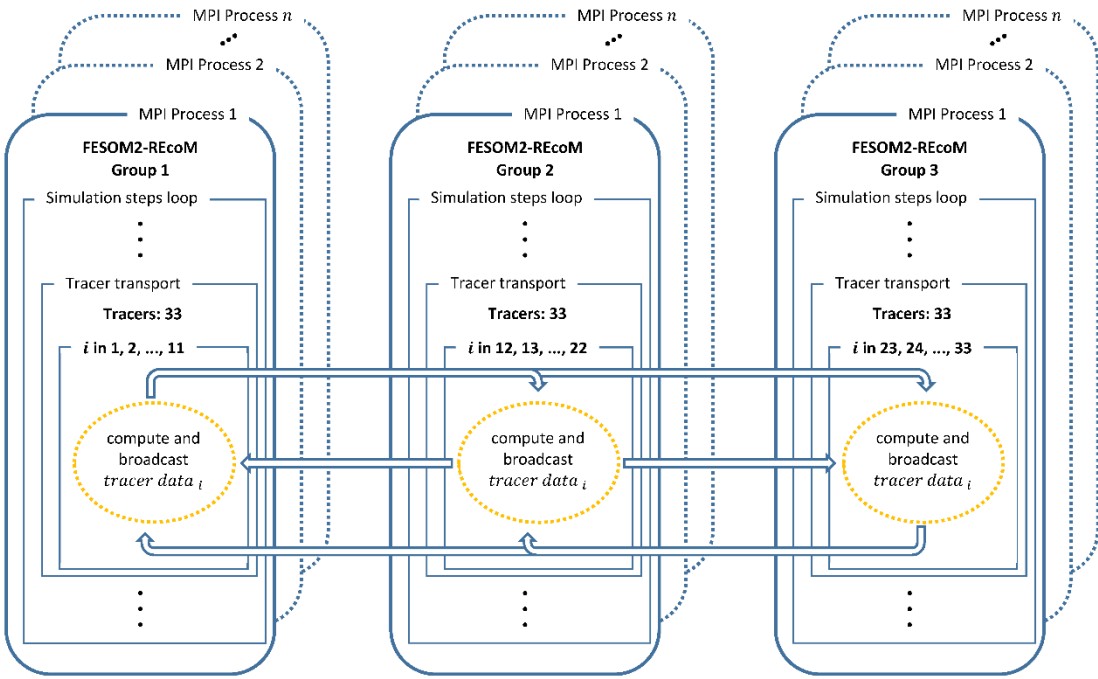

**Figure 2.** Exploiting parallelism at hotspot.

For MPI process 1 of groups 1 to 3 Fig. 2 shows the uniform splitting of the index range over 33 tracers, performing 11 iterations in each group. For the remaining MPI processes 2 to $n$, this splitting is done in exactly the same way due to the underlying domain decomposition, but for the sake of clarity the corresponding horizontal communication is not shown in Fig. 2. At the current state of the implementation, a synchronization step to broadcast partial results (referred to as $tracer\ data_i$) between groups is performed at the end of each iteration. The computation and communication phases could thus be overlapped very easily by using non-blocking MPI calls. On the avoidance of synchronization overhead, it will be beneficial if the computational effort for the different iterations is about the same. In this case benchmarks have shown that slightly better speedups are achieved with the overlap of the two phases than by omitting the synchronization step and broadcasting blocks of 11 partial results after the loop. Applied to the general case of a loop with hotspot character, larger differences in computational effort for different iterations can lead to load imbalance problems that could be addressed with a more sophisticated scheduling strategy such as work-stealing. Such strategies have not yet been examined for MULTIEXECMPI.

To determine the data structures relevant for $tracer\ data$, simply those were picked out in the FESOM2-REcoM source code that are potentially write-accessed in the loop over all tracers and are reused outside the loop. Basically, this is very similar to the procedure for a parallelization of the loop with OpenMP, where these data structures would have been declared



as *shared*. A deeper analysis of numerical computations or of already existing MPI communication structures was not necessary for this. This shows how a simulation model can largely be considered as a black box with MULTIEXECMPI. Nevertheless, a performance gain can still be achieved with a deeper analysis, if applicable. With the loop over all tracers, for example, tracer-specific output is also generated especially for writing restart files. It was therefore obvious not to merge this

output after each tracer loop iteration, and that in each simulation step, but to initially buffer it in the related groups and only merge it immediately before writing the restart files, which is comparatively rarely done. Benchmarks have shown that this is advantageous.

Figure 3 shows for FESOM2-REcoM, which is additionally coupled with ECHAM6 via OASIS, how data are exchanged between the models at the coupling step.

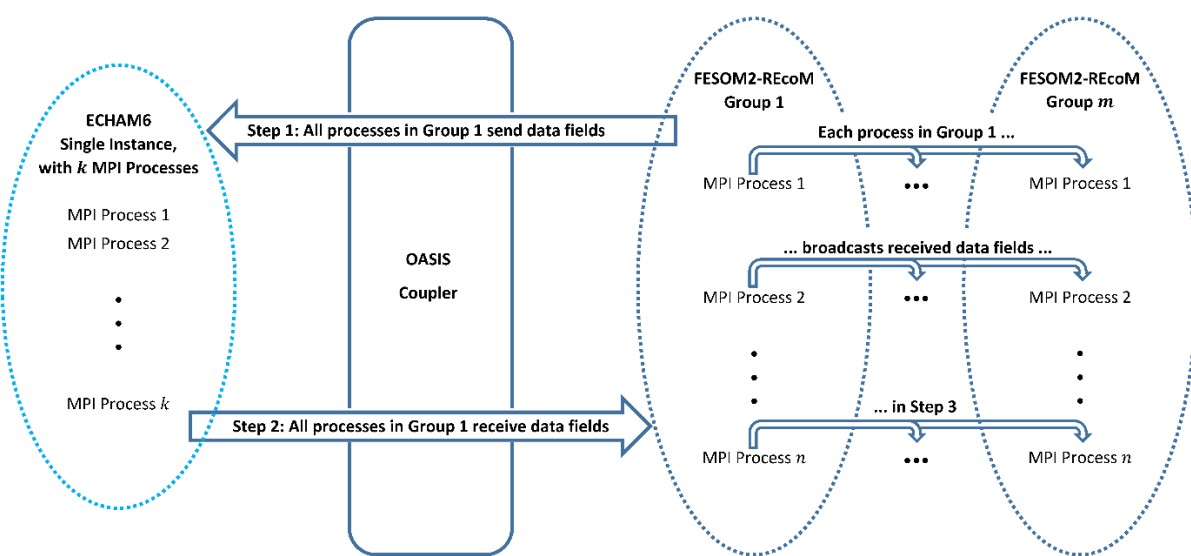


**Figure 3.** Data exchange at coupling step.

As with the writing of the result and restart files, group 1 has a special role, in which it initially behaves exactly like the single instance of the original implementation of FESOM2-REcoM. In a first substep, all data fields defined for the coupling are sent from the MPI processes in group 1 to ECHAM6 and in a second substep, all data fields sent from ECHAM6 are

received by the MPI processes in group 1. In a third substep, which is new for MULTIEXECMPI, data received from ECHAM6 are broadcasted to the remaining groups 2 to $m$, so that the MPI processes of different groups can subsequently continue their computations independently of each other. The additional substep can limit the parallelism at the coupling step depending on the effort for the broadcast of the data fields. Here an improvement is conceivable, for which ECHAM6 as well is executed multiple times in groups. The data fields between an ECHAM6 group $EG_i$ and a FESOM2-REcoM group $FG_i$ with the same $i$

in each case then only need to be exchanged with substeps 1 and 2. However, since a corresponding number of additional cores





is required for the additional ECHAM6 groups, this seems to make sense if at least one hotspot can also be parallelized with MULTIEXECMPI in ECHAM6. So far, this is still an open question.

### 3.1 Pitfalls in achieving bit-identical results

Parallelizations of numerical simulation models based on domain decomposition typically deliver deterministic simulation
results. Especially in the field of earth system modeling it is desirable to achieve bit-identical results when repeating simulation runs – also on different cluster systems – in the sense of reproducibility (see also Li et al., 2016). Within the same cluster system, reproducibility can also be very useful for checking that no errors have slipped in when changes are made to the source code. For example, a simple refactoring of the source code of a simulation model, which will have no influence on the results (e.g. a renaming of variables), will have been successful if with the refactored source code bit-identical results are obtained
again. After parallelizing a loop with MULTIEXECMPI, bit-identical results are also expected if there are no data dependencies between different iterations of the loop.

As part of the ongoing development of the biogeochemical model at the Alfred-Wegener-Institut (AWI), the calculation of a global sum to handle benthos fluxes was added to the loop over all tracers, which essentially resulted in data dependencies. The data dependencies could be easily avoided with the partitioning of the associated global data structures into tracer-specific
local data structures for buffering partial sums and for later calculation of the total sum, so that they may be considered neglectable. Even if both types of calculating the global sum are equivalent from a mathematical point of view, miniscule differences can arise due to the different order in which the associated floating-point operations are performed and, as a consequence, bit-identical results are usually no longer obtained. The effort for code review and testing was therefore considerably increased, since it was no longer possible to distinguish whether non-bit-identical results indicated an error in the
implementation, or whether they were merely the expected rounding differences. Nevertheless, bit-identical results can be obtained for repeated simulation runs of the same MULTIEXECMPI variant of FESOM2-REcoM and also for different number of groups, since aggregation of the tracer-specific partial sums immediately after the loop is always done in the same (ascending) order. In comparison, with a parallelization on the basis of OpenMP the concurrent shared memory accesses for the calculation of the global sum could at first be protected within a critical section (or something similar). Furthermore,
OpenMP potentially supports deterministic reduce operations to calculate global sums. With the underlying concurrency of the threads, however, no bit-identical results would be expected here either, since partial sums are already calculated within a single iteration in each individual thread. Such and similar problems are widely encountered in the domain of climate models. In this context, Geyer et al. (2021) propose to analyze numerical experiments also against a stochastic background.

### 3.2 Efficiency considerations

For a parallelization based on a domain decomposition, sooner or later a point will be reached at which an appropriate speedup can no longer be achieved with adding of further MPI processes. A major reason for this is that when the entire region is divided into smaller and smaller sub-domains, the effort for communication and synchronization during halo exchange



becomes large in relation to the simultaneously decreasing computational effort for a simulation step of a sub-domain. In addition, the maximum number of usable MPI processes may also be limited simply by the fact that a (small) region cannot be further divided into even smaller parts anyway. At the latest when such a point is reached, a domain decomposition can be combined with MULTIEXECMPI as a further independent parallelization approach.

Achieving adequate efficiency with MULTIEXECMPI requires that parallelizable hotspots of a simulation model make up a large part of the total computing time of the simulation runs. Performance engineering techniques such as benchmarking and profiling are appropriate to assess this. For FESOM2-REcoM, some early profiling experiments have shown that tracer transport computations make up about 80 % of the total computing time of a simulation run. Despite this remarkable percentage, the maximum achievable speedup remains rather limited. With an assumed remaining fraction of 20 %, which is not reduced with MULTIEXECMPI, the speedup 5 results as an upper limit even with a maximally good parallelization of the hotspot and the use of an arbitrary number of cores (see also remarks on Amdahl's Law, e.g., in Gustafson, 2011).

For parallelizing a loop with MULTIEXECMPI there must be an appropriate relation between the computational and the communication effort, which is the more favorable the more can be computed in one iteration and the less data must be exchanged for the merge operations between the groups. With an increasing number of groups the communication effort rises and therefore it will become difficult to still achieve an appropriate ratio to the computational effort. Here a parallelization on the basis of OpenMP is advantageous, since the additional communication effort is not necessary. In efficiency considerations, on the other hand, there are no principle advantages for an OpenMP approach, because in the phases in which, for $m$ groups, the MPI processes perform redundant computations in $m - 1$ groups outside the parallelized loops, the additional $m - 1$ threads of an MPI process would be idle in an analogous sense. The fact that the power consumption of a cluster node on which several cores perform redundant computations for a certain fraction of the total runtime may be greater than if these cores were idle for a comparable fraction of the total runtime is not considered for this comparison. In the original implementation of FESOM2-REcoM, the tracer transport computation has such a dominant percentage of the total runtime that it will be difficult to identify other loops as hotspots.

## 4 Experimental results

Presented below are the benchmark results determined with controlled experiments for extensions of FESOM2 to include a Lagrangian-based iceberg model (Rackow et al., 2017; Ackermann et al., 2022) (hereafter referred to as FESOM2-Iceberg) and a biogeochemical model (Gurses et al., 2021) (hereafter referred to as FESOM2-REcoM). The iceberg and biogeochemical models are developed at AWI and they have not been modified with respect to their scientific computations as part of the accelerations. The first benchmarks were performed on the Ollie cluster at AWI and the Mistral cluster at DKRZ. Further benchmarks were then performed on the Levante cluster, as successor of the Mistral cluster, at DKRZ. In the interpretation of the results, this will be briefly discussed at the relevant points. To deal with the problem of outliers in benchmark results (e.g.



due to the use of shared cluster resources like network or I/O devices) runs were repeated several times and the median was

chosen.

### 4.1 FESOM2 plus icebergs

Before presenting the results for parallelizing the FESOM2-Iceberg model in more detail, two other approaches for speeding up the model will be briefly discussed, which have been implemented previously and which can be combined with the parallelization based on running FESOM2-Iceberg multiple times to split and accelerate the loop over all icebergs. These two

approaches are based on the observation that the ocean/ice model in FESOM2 and the iceberg model are rather loosely coupled.

One of the approaches to speed up the computation (implemented at AWI) is thus to use different simulation time steps (hereafter also referred to as steps for short) for the model components, such as 4 steps in the ocean/ice model per each step in the iceberg model. For the other approach, $n$ steps in the ocean/ice model can additionally be computed overlapped with their corresponding step in the iceberg model in the sense of asynchronously coupled models. In the ideal case, $n$ steps in the

ocean/ice model require the same computational effort as a single step in the iceberg model and with this optimal overlap an additional speedup of 2 can be achieved relative to sequential coupling. The asynchronous coupling, also implemented as part of the PalMod project at DKRZ (PalMod, 2022), is very similar to the extension of ECHAM6 for overlapping the computation of the atmospheric and radiation components (Heidari et al., 2021). With the variation of the time step ratios and with the asynchronous coupling the acceleration of the model is at the cost of a slightly reduced accuracy of the simulation results.

With the additional combination of these two approaches with MULTIEXECMPI redundant computations are accepted for a further acceleration. This shows the importance of runtime reductions in general.

The following benchmark results are based on representative input data provided by AWI for the controlled experiments. For results with good relevance to practice, the FESOM2 iceberg model was additionally coupled with ECHAM6 for all experiments, in accordance with its major use as a subcomponent of the so-called AWI-ESM (Earth System Model). Since

ECHAM6 itself is not accelerated during the experiments, it can be assumed that the speedups for the fully coupled AWI-ESM in this way will be somewhat lesser for the parallelization approaches given above than if similar experiments had been performed with a stand-alone variant of FESOM2-Iceberg. To mitigate this somewhat, it seems fair to include the nodes used by ECHAM6 for efficiency considerations.

Table 1 first shows how the asynchronous coupling of the ocean/ice and iceberg model performs for 1 and 10,000 artificially

– with respect to reality – generated icebergs. (For production runs, it is planned to perform experiments with up to 100,000 icebergs in the future.)



**Table 1**. AWI-ESM "ECHAM6+FESOM2-Iceberg" speedups – 1 vs. 10,000 icebergs.

**Asynchronous Coupling of FESOM2 with Iceberg Model**

**Setup: core2-Mesh, Simulation Period Jan. 1–31, 1850, Ollie Cluster**

| Icebergs | Async. Iceberg Coupling | ECHAM6 + FESOM2-I. Nodes | Runtime [s] | Speedup | FESOM2-I. Efficiency | Total Efficiency |
|---|---|---|---|---|---|---|
| 1 | no | 12 + 8 | 215 | 1.00 | 100.00 % | 100.00 % |
| 1 | yes | 12 + 16 | 190 | 1.13 | 56.50 % | 80.71 % |
| | | | | | | |
| 10,000 | no | 12 + 8 | 299 | 1.00 | 100.00 % | 100.00 % |
| 10,000 | yes | 12 + 16 | 189 | 1.58 | 79.00 % | 112.86 % |

The asynchronous coupling of the ocean/ice and iceberg-model was implemented with OpenMP. For the experiments with "yes" in the corresponding column, each FESOM2-Iceberg MPI process contains two threads, one for the ocean/ice and one for the iceberg computations, thus the number of nodes is doubled. The ratio of ocean/ice steps per iceberg step was set to 2:1 for all experiments, so that for the asynchronous case and the larger number of 10,000 icebergs a good compromise between the accuracy of the simulation results and a good overlap of ocean/ice steps with an iceberg step is obtained. The number of nodes used for ECHAM6 was determined with preliminary experiments in the sense of tuning. It is chosen to be 12 nodes, because experiments showed that a doubling to 24 nodes would barely reduce the runtimes of the fully coupled model any further. This also applies to the cases where FESOM2-Iceberg is accelerated and for which one would not actually expect this. With the elimination of the load imbalances between the accelerated FESOM2-Iceberg model and ECHAM6, which can be assumed as the reason for this, it should be possible to reduce the runtimes even more. However, in order to keep the effort for the experiments manageable, this was not attempted within the scope of this paper.

For the experiments with 1 iceberg, only a small speedup can be achieved with asynchronous coupling, as expected, since in the iceberg model besides some basic overhead for the coupling there is hardly any effort for computations. In the experiment with 10,000 icebergs, when doubling the nodes for FESOM2-Iceberg from 8 to 16 a speedup of 1.58 is achived with an efficiency of 79 %. If the 12 nodes used in the experiments for ECHAM6 are additionally taken into account, a total efficiency $E_T = 1.58 \,/(28/20)$ of almost 113 % results, which could even be assessed in some sense as a superlinear speedup with the considerations on fair efficiencies above.

The new approach to start FESOM2-Iceberg multiple times in groups can be combined with the asynchronous approach to achieve a twofold parallelization scheme (on top of the domain decomposition based parallelization), whereby the speedups can be achieved independently of each other. Figure 4 shows, using the example of an ocean/ice and iceberg step $n$, how the computation time can be reduced in this way.



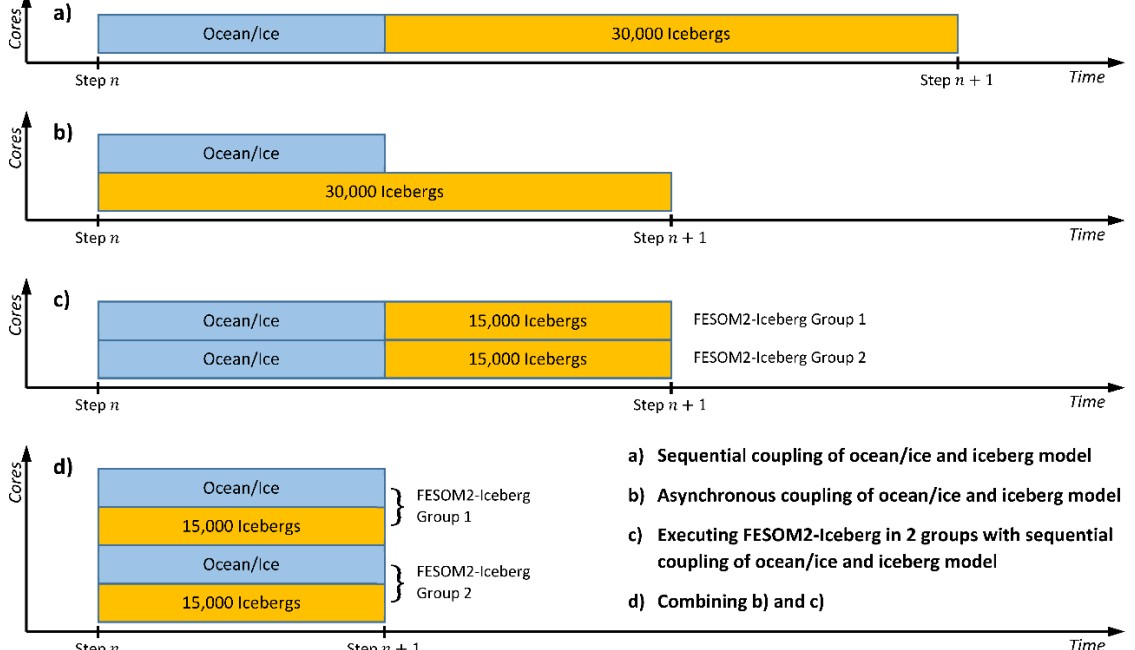

**Figure 4.** Twofold parallelization scheme for FESOM2-Iceberg (ideal conditions assumed).

For the sake of simplicity, overheads for synchronization and communication in the sense of ideal conditions are neglected for Fig. 4, and FESOM2-Iceberg is not coupled with ECHAM6. The ratio of ocean/ice steps per iceberg step is taken as 1:1 and the computational effort for an iceberg step with 30,000 icebergs in the sequential case is assumed to be twice as large as for an ocean/ice step. In the example, the asynchronous approach in b) cannot yet fully show its potential due to the non-optimal overlap of the ocean/ice and iceberg computations, which limits the speedup to 1.5 compared to the sequential variant in a). When executing FESOM2-Iceberg in 2 groups in c), the computation time for the iceberg step can be reduced to half under the ideal conditions. With the non-accelerated ocean/ice step, however, the speedup is also limited to 1.5. With the combination of the two approaches in d), a speedup of 3 can be achieved with a quadrupling of the computing power.

Table 2 contains representative benchmark results for parallelizing the loop over 30,000 icebergs using the new approach to start FESOM2-Iceberg multiple times in groups.



**Table 2.** AWI-ESM "ECHAM6+FESOM2-Iceberg" speedups – variation of step ratio.

**Iceberg Loop Parallelization**

**Setup: core2-Mesh, Simulation Period Jan. 1–31, 2001, 30,000 Icebergs, Mistral Cluster**

| FESOM2-I. Groups | Async. Iceberg Coupling | ECHAM6 + FESOM2-I. Nodes | Ocean/Ice Steps per Iceberg Step | Runtime [s] | Speedup | FESOM2-I. Efficiency | Total Efficiency |
|---|---|---|---|---|---|---|---|
| 1 | no | 12 + 8 | 1 | 987 | 1.00 | 100.00 % | 100.00 % |
| 1 | yes | 12 + 16 | 1 | 745 | 1.32 | 66.00 % | 94.29 % |
| 2 | no | 12 + 16 | 1 | 714 | 1.38 | 69.00 % | 98.57 % |
| 2 | yes | 12 + 32 | 1 | 578 | 1.71 | 42.75 % | 77.73 % |
| | | | | | | | |
| 1 | no | 12 + 8 | 4 | 531 | 1.00 | 100.00 % | 100.00 % |
| 1 | yes | 12 + 16 | 4 | 426 | 1.25 | 62.50 % | 89.29 % |
| 2 | no | 12 + 16 | 4 | 420 | 1.26 | 63.00 % | 90.00 % |
| 2 | yes | 12 + 32 | 4 | 342 | 1.55 | 38.75 % | 70.45 % |

The table shows the results for starting FESOM2-Icebergs as a single group, which corresponds to the way FESOM2-Iceberg was originally used when coupled with ECHAM6, and for starting FESOM2-Icebergs in 2 groups. Experiments with the start of 3 or more FESOM2-Iceberg groups are not included because preliminary tests have shown that no further speedup can be achieved in this case. The main reason is that the communication overhead for reestablishing the same internal state in all FESOM2-Iceberg groups is too much of an issue when merging partial results while iterating in the loop over the icebergs. For ECHAM6, 12 nodes are used again, since a doubling of the nodes would barely reduce the runtimes of the fully coupled model further. As with the corresponding considerations for Table 1, it can be assumed that the elimination of the load imbalances between the accelerated FESOM2-Iceberg model and ECHAM6 would reduce the runtimes even more.

Note that the experiments for the first two rows in Table 2 are similar to the experiments for the last two rows in Table 1, so that one could expect a superlinear speedup or a total efficiency of more than 100 % in row 2 of Table 2. However, a direct comparison of results in different tables is not always possible. The main reason for this is that both the Iceberg model and REcoM have been evolved at AWI over the entire period of the experiments (totally independent of their parallel extension), which also regularly resulted in changes with respect to practice-relevant setups (mesh resolution, simulation year, simulation steps per day, input data, ...). Even though the simulation models for the parallelization approach presented here are essentially a black box, this can still result in smaller impacts on runtime behavior. In addition, the experiments for Table 1 and Table 2 here also differ in the number of icebergs and the ocean/ice steps per iceberg step ratios.

With a larger number of ocean/ice steps per iceberg step, an additional speedup can be achieved, which is reflected in the experiments for Table 2. In addition to a step ratio of 1:1, the ratio of 4:1 was therefore chosen, which represents an appropriate compromise of additional speedup with a slightly reduced accuracy of the simulation results. However, if the ratio would be



increased even further, a problem may be that the computational effort for an iceberg step is already reduced to such an extent that it will essentially be almost impossible to achieve a further significant speedup with an overlapped computation with the ocean/ice steps. For the step ratio of 1:1, the two parallel approaches achieve approximately the same speedup of 1.32 and 1.38, respectively. With the combination of the two independent approaches, a speedup of 1.71 can be achieved, but for this

the number of nodes must be doubled twice. The "FESOM2-Iceberg efficiency" thus drops to nearly 40 %. If the nodes used for ECHAM6 are additionally taken into account, the efficiency is at least still 77.73 %.

With the step ratio of 4:1 and without the use of another parallel approach, the runtime can first be reduced from 981 to 531 seconds in comparison with the corresponding experiment with the step ratio of 1:1. For the two parallel approaches, the speedups of 1.25 and 1.26 are again about the same, but somewhat lower than the corresponding results for the 1:1 ratio of the

steps. For the speedup of the asynchronous coupling it can be assumed that the 4:1 ratio already has a somewhat disadvantageous effect on a suitable overlap of the ocean/ice and iceberg model steps, as it was described above for the extreme case of a very large ratio of the steps. When FESOM2-Iceberg is started in 2 groups, the ocean/ice model itself is not accelerated, so that its part of the total runtime increases accordingly for a 4:1 ratio of steps. Despite the same assumed relative speedups of the iceberg model for step ratios of 1:1 and 4:1, this increase has a correspondingly disadvantageous effect on the

speedup when total runtimes are considered. Accordingly, the speedup of 1.55 with the combination of the two approaches also turns out to be somewhat lower than before. The total efficiency is still slightly above 70 %.

Table 3 contains representative benchmark results for the parallelization as achieved after adapting the models to the Levante cluster. The number of icebergs has been increased to about 40,000.

**Table 3.** AWI-ESM "ECHAM6+FESOM2-Iceberg" speedups.

**Iceberg Loop Parallelization**

**Setup: core2-Mesh, Simulation Period Jan. 1–31, 1850, ≈ 40,000 Icebergs, Levante Cluster**

| FESOM2-I. Groups | Async. Iceberg Coupling | ECHAM6 + FESOM2-I. Nodes | Runtime [s] | Speedup | FESOM2-I. Efficiency | Total Efficiency |
|---|---|---|---|---|---|---|
| 1 | no | 2 + 2 | 617 | 1.00 | 100.00 % | 100.00 % |
| 1 | yes | 2 + 4 | 417 | 1.48 | 74.00 % | 98.67 % |
| 2 | no | 2 + 4 | 471 | 1.31 | 65.50 % | 87.33 % |
| 2 | yes | 2 + 8 | 309 | 2.00 | 50.00 % | 80.00 % |

The ratio of ocean/ice timesteps per iceberg timesteps was set to 2:1 as a compromise between the accuracy of the simulation results and its impact on overall model speedup. On Levante, similar domain decompositions were used for ECHAM6 and



FESOM2-Iceberg as for the previously described experiments on Mistral. Since a node of Levante has 128 cores[1] and a node of Mistral has only 36 cores, the number of nodes used here are correspondingly smaller. For ECHAM6, 2 nodes were used, since a doubling would again barely reduce the runtimes of the fully coupled model further.

The speedup achieved with asynchronous coupling of the iceberg model is 1.48. This is slightly better than the comparable speedup of 1.32 from Table 2 on Mistral, which was obtained there for a step ratio of 1:1. It can be assumed that in the experiments on Levante the larger number of icebergs (about 40,000 compared to 30,000 on Mistral) had a beneficial effect on the overlap of ocean/ice and iceberg steps. In addition, small differences between the two cluster systems can be argued to be a reason for the difference in speedups. Furthermore, fewer nodes are used for the experiments on Levante than on Mistral,

so that on Levante a considerably larger fraction of the MPI communication can be performed particularly efficiently via shared memory. When FESOM2-Iceberg is started in 2 groups, the speedup of 1.31 is about in between the comparable speedups of 1.26 and 1.38 from Table 2 on Mistral, which were determined there for the step ratios of 4:1 and 1:1. For the speedup of 2 when combining the two approaches, the efficiency on Levante is still 80 % when considered fairly as $E_F = 2/(10/4)$, although only 2 ECHAM6 nodes can be additionally considered for this.

## 4.2 FESOM2 plus REcoM

Some profiling experiments for adding the biogeochemical model to FESOM2 have shown that about 80 % of the total time of a simulation run is spent on the tracer transport computations. This included 33 tracers (e.g. dissolved inorganic carbon and alkalinity for the carbonate system, the macronutrients dissolved inorganic nitrogen and silicic acid and the trace metal iron – to name a few), according to practice-based configurations for representative input data. The key hotspot that could be

identified very quickly was a compute-intensive loop over these tracers that is executed for each time step of the simulation. Luckily, the tracer transport computations can be performed independently for all loop indices, which is an ideal basis for loop parallelization.

In a first approach, the tracer loop was parallelized based on OpenMP. Although the OpenMP implementation basically required only a number of simple and essentially similar changes, some effort was nevertheless required, especially for the

handling of MPI calls contained within the tracer loop, and a number of data structures in quite a few source files had to be modified. Among the REcoM developers this led to the desire for an alternative solution that would avoid this variety of small changes and the additional complexity of a hybrid approach. According to the developers, such an alternative solution, based on pure MPI, would also be more likely to be accepted by the FESOM2 community (e.g. for a later merge into a main development branch) if the source code only needed to be changed minimally. This was the main reason for searching for

alternatives and developing the MULTIEXECMPI approach. Anyway, the start of FESOM2-REcoM in groups for the parallelization of the loop over all tracers is in this case in principle independent of the already underlying parallelization of

---

[1] If in the following "cores" is used in the text, "phys. cores" are meant. For the simulation models used in this paper, the use of hyper-threaded cores would generally have resulted in somewhat longer runtimes. For some considerations on phys. vs. hyper-threaded cores, see for example the remarks in (Himstedt et al., 2019, p. 63 ff.).





the same loop on the basis of OpenMP. It was therefore possible and obvious to also perform some experiments combining the two methods in the sense of a twofold parallel approach, although they are actually rather implementation alternatives.

Table 4 shows which speedups can be achieved with the two methods on Mistral depending on the number of cores used.

**Table 4.** FESOM2-REcoM Speedups – MPI based groups vs. OpenMP threads.

**Twofold Tracer Loop Parallelization**
**Setup: pi-Mesh, Simulation Period Jan.–Dec., 1948, 33 Tracers, Mistral Cluster**

| FESOM2-R. Groups | Threads per MPI Process | Total Cores | Runtime Tracers [s] | Speedup Tracers | Runtime [s] | Speedup | Efficiency |
|---|---|---|---|---|---|---|---|
| 1 | 1 | 36 | 740 | 1.00 | 943 | 1.00 | 100.00 % |
| 1 | 2 | 72 | 409 | 1.81 | 589 | 1.60 | 80.00 % |
| 2 | 1 | 72 | 431 | 1.72 | 631 | 1.49 | 74.50 % |
| 1 | 3 | 108 | 299 | 2.47 | 476 | 1.98 | 66.00 % |
| 3 | 1 | 108 | 328 | 2.26 | 523 | 1.80 | 60.00 % |
| 1 | 6 | 216 | 228 | 3.25 | 460 | 2.05 | 34.17 % |
| 6 | 1 | 216 | 257 | 2.88 | 453 | 2.08 | 34.67 % |
| 2 | 3 | 216 | 173 | 4.28 | 357 | 2.64 | 44.00 % |
| 3 | 2 | 216 | 197 | 3.76 | 369 | 2.56 | 42.67 % |

The column "Threads per MPI Process" contains the number of threads that were available within each MPI process for the OpenMP parallelization of the loop over all tracers. The column "Runtime Tracers" shows the part that the tracer transport computation had in the corresponding total runtime. For the base variant of FESOM2-REcoM (with 1 group and 1 thread per MPI process) this is $740s/943s \approx 78$ %, which corresponds well to the ratio of 80 % from the early profiling experiments.

The entries in the "Speedup Tracers" column, whose calculation is based on the entries in the "Runtime Tracers" column, give an indication of the potential upper limits for the achievable speedups. For the experiments with 72 cores (i.e. using the number of 2 threads per MPI process or 2 groups) good relative speedups of 1.81 and 1.72 and somewhat lower absolute speedups of 1.60 and 1.49 are achieved. The corresponding efficiencies of 80.00 % and 74.50 % appear to be quite adequate.

With increasing parallelization, the fraction of time that the tracer transport computation has in the total runtime decreases.

For the experiments where 216 cores were used, the fraction of the total runtime can be reduced from about 78 % for the base variant to slightly below 50 % in two cases: 1 group and 6 threads per MPI process with $228s/460s \approx 49.6$ % or 2 groups and 3 threads per MPI process with $173s/357s \approx 48.5$ %. On the other hand, this makes the remaining part of the total runtime, for all non-accelerated computations outside the loop over all tracers, more of a factor. If one assumes that even with a maximally good parallelization of the loop this remaining part still is 20 % of the total runtime, this would limit the speedup

to 5 (see also Sect. 3.2 with the reference to Amdahl's Law).



For the number of 6 FESOM2-REcoM groups or 6 threads per MPI process, a rather limited scalability becomes apparent with efficiencies of less than 35 % for both approaches. It is interesting to note when looking at the results for the remaining two experiments with 216 cores that combining the two methods achieves indeed slightly better speedups (2.64 or 2.56) than using each approach individually (2.05 or 2.08). Since both approaches are independent of each other, an overall speedup could be estimated by multiplying the two speedups obtained separately. Even though the total speedups in the table are somewhat lower than the values calculated in this way, they are an indication that the combination of independent parallelization methods will allow the benefits of often underlying orthogonality – as in this case between MPI and OpenMP – to be exploited in this way. However, the hybrid solution will not be convenient for being applied to other problems, since it is especially the complexity associated with an additional OpenMP implementation that shall be avoided with executing an MPI application multiple times in groups. An in-depth analysis of the observation was therefore omitted as well.

Table 5 shows which speedups can be achieved depending on the multiple start of FESOM2-REcoM in groups on the two cluster systems at DKRZ. It should be briefly noted here that a direct comparison with the results in Table 4 is not possible. As has been pointed out in connection with the results of the Iceberg model, also REcoM has been evolved at AWI over the entire period of the experiments, which also regularly resulted in changes with respect to practice-relevant setups (mesh resolution, simulation year, simulation steps per day, ...). This can result in smaller impacts on runtime behavior. Additional experiments for the OpenMP parallelization of the loop over all tracers, which is not in the focus of this paper, were not performed.

**Table 5.** FESOM2-REcoM speedups – Mistral vs. Levante.

**Tracer Loop Parallelization based on FESOM2-REcoM Groups**
**Setup: core2-Mesh, Simulation Period Jan.–Dec., 2000, 33 Tracers**

| FESOM2-R. Groups | Cluster | Nodes | Runtime Tracers [s] | Speedup Tracers | Runtime [s] | Speedup | Efficiency |
|---|---|---|---|---|---|---|---|
| 1 | Mistral | 8 | 8549 | 1.00 | 9803 | 1.00 | 100.00 % |
| 2 | Mistral | 16 | 4605 | 1.86 | 5746 | 1.71 | 85.50 % |
| 3 | Mistral | 24 | 3087 | 2.77 | 4259 | 2.30 | 76.67 % |
| 4 | Mistral | 32 | 2601 | 3.29 | 3802 | 2.58 | 64.50 % |
| 1 | Levante | 8 | 3448 | 1.00 | 3776 | 1.00 | 100.00 % |
| 2 | Levante | 16 | 1850 | 1.86 | 2263 | 1.67 | 83.50 % |
| 3 | Levante | 24 | 1358 | 2.54 | 1669 | 2.26 | 75.33 % |
| 4 | Levante | 32 | 1213 | 2.84 | 1523 | 2.48 | 62.00 % |

A single FESOM2-REcoM group uses 8 nodes on the respective cluster system, which corresponds to a total of 1024 cores on Levante (with 128 cores per node) and 288 cores on Mistral (with 36 cores per node). The performance of a single core can





be assumed to be about the same on both cluster systems. For experiments using the same number of nodes on both cluster systems, Levante would be expected to have a speedup of $128/36 \approx 3.6$ compared to Mistral. In fact, the speedups are significantly lower, for example $9803s/3776s \approx 2.6$ for the experiments with 1 group. The first reason for this can be attributed to the fact that the Levante cluster had only been in operation for a short time and there is presumably still potential

for optimization for some MPI parameters or compiler options. For instance, OpenMPI is used on Mistral and IntelMPI on Levante. Furthermore, the Levante architecture is based on AMD CPUs while Mistral architecture is based on Intel CPUs, which can have a noticeable impact on building efficient applications. As a main reason, however, it can be assumed that the larger number of MPI processes used for the domain decomposition in the experiments on Levante already leads to a slightly less acceptable strong scaling behavior of FESOM2-REcoM. For the relative speedups that are additionally achieved with

MULTIEXECMPI independently of the underlying domain decomposition, on the other hand, it can be assumed that this larger number of MPI processes does not cause a disadvantageous effect in the experiments on Levante. Overall, the comparison of the results between Mistral and Levante therefore shows, as expected, that despite the different absolute runtimes roughly the same speedups are achieved on both systems.

Figure 5 uses the Simulated Years Per Day (SYPD) performance metric based on the speedups in Table 5 to show how the

tracer loop parallelization scales on the two cluster systems.

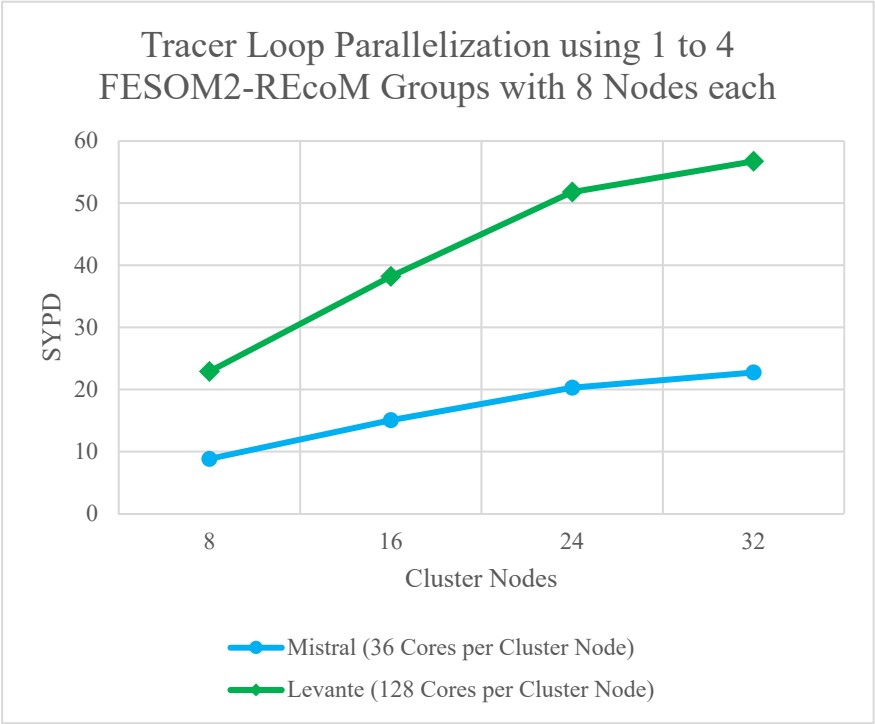

**Figure 5.** FESOM2-REcoM Simulated Years Per Day (SYPD) performance.




For both cluster systems, the uniform increase in SYPD when starting FESOM2-REcoM in up to 3 groups is well shown. Then, when starting FESOM2-REcoM in 4 groups, using 32 nodes each, the two curves clearly flatten. The overall stronger
increase in the Levante curve does not mean that the approach scales better on Levante than on Mistral. It is due to the fact that the number of nodes used are plotted for a common x-axis, but a single Levante node has significantly more cores than a Mistral node.

Table 6 shows for the Levante cluster what speedups are still achieved when FESOM2-REcoM is coupled to ECHAM6 as a subcomponent of the AWI-ESM. As with the analogous coupling of the FESOM2-Iceberg model with ECHAM6 for the
experiments presented in Sect. 4.1, ECHAM6 itself is not accelerated. Therefore, it seems fair to mitigate somewhat the disadvantages this has on the overall speedup by including the nodes used by ECHAM6 in efficiency considerations.

**Table 6.** AWI-ESM "ECHAM6+FESOM2-REcoM" speedups.

**Tracer Loop Parallelization based on FESOM2-REcoM Groups**

**Setup: core2-Mesh, Simulation Period Jan. 1–31, 2000, 33 Tracers, Levante Cluster**

| FESOM2-R. Groups | ECHAM6 + FESOM2-R. Nodes | Runtime Tracers [s] | Speedup Tracers | Runtime [s] | Speedup | FESOM2-R. Efficiency | Total Efficiency |
|---|---|---|---|---|---|---|---|
| 1 | 4 + 8 | 167 | 1.00 | 298 | 1.00 | 100.00 % | 100.00 % |
| 2 | 4 + 16 | 88 | 1.90 | 217 | 1.37 | 68.50 % | 82.20 % |
| 3 | 4 + 24 | 75 | 2.23 | 207 | 1.44 | 48.00 % | 61.71 % |

A single FESOM2-REcoM group uses 8 nodes (with a total of 1024 cores). The number of nodes used for ECHAM6 is again chosen such that even if doubled, that would barely reduce the runtimes of the fully coupled model any further. For the
use of 2 FESOM2-REcoM groups, the speedup of 1.90, which results with respect to the tracer loop, can be estimated as quite good. As expected, the overall speedup of 1.37 is significantly lower. In a fair view, i.e. with additional consideration of the nodes used for ECHAM6, the efficiency is still 82.20 %. With the use of 3 FESOM2-REcoM groups, the speedup can still be slightly increased, but the FESOM2-REcoM efficiency drops to below 50 % and even with a fair consideration, the efficiency is only just above 60 %.

**5 Conclusions and future work**

In various scientific fields it is not uncommon to consciously accept redundant computations for parallel approaches to reduce the time-to-solution, as it was shown with some examples in Sect. 2. However, for climate models based on a domain decomposition this has not been evaluated so far. In this paper, the two examples of FESOM2-Iceberg and FESOM2-REcoM have shown how such ideas can be successfully applied to accelerate climate models as well. In this context, it was an attractive
idea to consider the existing simulation application largely as a black box and to implement the additional parallelization of



compute-intensive loops with MULTIEXECMPI as a second independent level on top of the domain decomposition. With the steadily increasing number of cores per cluster node it can become increasingly difficult to utilize them merely on the basis of a domain decomposition. If a region can no longer be (meaningfully) broken down into smaller parts, the point is reached at which a combination with MULTIEXECMPI can utilize additional cores and achieve a further speedup.

Experiments with 30,000 icebergs have shown for the coupling of FESOM2-Iceberg with ECHAM6 that a speedup of 1.38 can be achieved with the parallelization of the loop over all icebergs with MULTIEXECMPI on the Mistral cluster when doubling the cores used for FESOM2-Iceberg from 288 to 576. In the additional combination with a parallel approach for asynchronous coupling of the ocean/ice and iceberg model and the further doubling of the cores used for FESOM2-Iceberg from 576 to 1,152, the speedup can be increased to 1.71. For experiments on the Levante cluster with about 40,000 icebergs, a speedup of

2.00 is achieved when quadrupling the cores used for FESOM2-Iceberg from 256 to 1024 and combining MULTIEXECMPI with the asynchronous coupling of the ocean/ice and iceberg model. The efficiency is about 80 %, taking into account the 256 cores used for ECHAM6. In the FESOM2-Iceberg experiments, no further speedup could be achieved with the execution of 3 or more FESOM2-Iceberg groups, because the communication overhead for reestablishing the same internal state in all groups is too much of an issue when merging partial results.

In the original implementation of FESOM2-REcoM, the tracer transport computations with the loop over all tracers make up a dominant percentage of the total computation time (about 80 % for 33 tracers). This loop was parallelized based on OpenMP as well as on MPI via MULTIEXECMPI. Overall, the comparison of the results shows a high degree of similarity between the two methods. Only minimal source code changes were necessary in each case to parallelize the loop. For OpenMP, these are essentially the changes for the private/shared differentiation of data structures and for handling concurrent MPI calls.

For MULTIEXECMPI, these are mainly the changes for the broadcasting of partial results of the loop iterations between the groups, whereby in this case computation and communication phases were overlapped. For the parallelization of a compute-intensive loop based on OpenMP, the potential of shared memory access represents an advantage if with MULTIEXECMPI significantly more data have to be transferred to merge partial results at a hotspot than was necessary for the loop over all tracers. An implementation based on MULTIEXECMPI in turn offers the advantage over the hybrid approach of avoiding

dependencies from certain OpenMP features (e.g. a performant support of MPI_THREAD_MULTIPLE) and additional complexity with respect to the use of the compiler, MPI and runtime environment. In addition, it can be assumed for MULTIEXECMPI that debugging is also less complex than for a hybrid application (even if this is very well supported with powerful debuggers like arm DDT (Distributed Debugging Tool) (Arm, 2022)).

For parallelizing the tracer loop with MULTIEXECMPI experiments show that about the same relative speedups can be

achieved on the Mistral and the Levante cluster. When quadrupling the nodes used for FESOM2-REcoM from 8 to 32, a speedup of 2.58 is achieved on the Mistral cluster and a speedup of 2.48 is achieved on the Levante cluster. For additional practical relevance of the results, FESOM2-REcoM was also coupled with ECHAM6. For the fully coupled model and when doubling the nodes used for FESOM2-REcoM from 8 to 16 for the use of 2 FESOM2-REcoM groups, experiments on Levante show a good relative speedup of 1.90 when considering the tracer loop speedup in isolation. With 1.37 the overall speedup for



the fully coupled model is lower, as expected. The associated efficiency, for which it seemed fair to include the 4 nodes used for ECHAM6, is about 82 %. With the use of 3 FESOM2-REcoM groups, the total speedup can be increased slightly to 1.44, but the total efficiency is then only just over 60 %. Which efficiencies still seem acceptable in practice will depend on the relevance of a further reduction of the time-to-solution for a given problem.

    With respect to the simplicity of MULTIEXECMPI, the speedups shown above seem reasonable for the small number of 2

and to some extent up to 4 groups used in the experiments. For an applicability to other simulation models it is essential that for the split of a compute-intensive loop no data dependencies exist between the statements of different iterations and that in relation to the computational effort of an iteration only few data have to be exchanged between the groups for the merge of partial results. However, even under maximally suitable conditions the maximum achievable speedup remains rather limited, which was explained with reference to Amdahl's Law in Sect. 3.2. The same applies, however, for a parallelization based on

OpenMP.

    Ideally, the (organizational) conditions in a project allow to accelerate a large simulation model, which has been developed as an MPI application for many years and to which a large number of developers have constantly added new functionalities after a deep analysis of the source code with sophisticated ideas. MULTIEXECMPI can rather be used when this ideal case is not given. But also for an already optimized simulation model it can be checked without special effort if a further acceleration

can be achieved with the independent level of parallelism of MULTIEXECMPI. Performance engineering techniques such as benchmarking and profiling and an analysis of the source code for data dependencies of compute-intensive loops are sufficient for this. For MPI calls, which may be contained within the loops, no additional (synchronization) effort is required anyway. This will have an overall positive impact on a good readability and maintainability of the source code, which as a side effect can facilitate the acceptance of the minimal code changes in the corresponding developer community.

**5.1 Future work**

At AWI, for the further development of the AWI-ESM, it is planned to merge the source code of the so far separately embedded iceberg and biogeochemical models into FESOM2. With respect to MULTIEXECMPI, no merge conflicts are to be expected, because the implementation for the startup phase, the communicator splitting, the coupling via OASIS, etc. is the same for FESOM2-Iceberg and FESOM2-REcoM.

One extension idea is to use the potential of the MPI processes additionally available with MULTIEXECMPI also outside of parallelized loops, for example for the parallel writing of result and restart files. Another idea is to evenly split expensive collective MPI calls between the groups to subsequently merge already aggregated partial results between them. It is quite conceivable that this could achieve a performance gain depending on the amount of data to be transmitted, available bandwidth, arrangement of MPI processes on the cluster nodes, etc.

In Sect. 3 it was described in the context of coupling FESOM2 (executed via MULTIEXECMPI) with ECHAM6 that for the data exchange an additional substep is required to broadcast the data received in FESOM2 group 1 from ECHAM6 to all other FESOM2 groups. Here, an interesting question is how the time-to-solution can be reduced if ECHAM6 as well is executed in

groups to avoid the additional substep, and whether ECHAM6 contains hotspots that are suitable for a parallelization with MULTIEXECMPI.

In addition, it seems particularly exciting to investigate further simulation models with regard to their suitability for parallelization with MULTIEXECMPI. This is especially true for models which, in contrast to the two models investigated so far, do not contain a single – quasi dominant – loop but several compute-intensive loops without data dependencies.

**Code availability**

FESOM2-Iceberg extended by the MULTIEXECMPI source code is available at https://doi.org/10.5281/zenodo.7835286. The
OASIS coupler extended by the MULTIEXECMPI source code is also included there. FESOM2-REcoM extended by the MULTIEXECMPI source code is available at https://doi.org/10.5281/zenodo.7835316.

**Author contribution**

Conceptualization and implementation of the MULTIEXECMPI approach presented in this paper as well as performing the benchmarks and analyzing the results was done by KH. FESOM2-Iceberg and FESOM2-REcoM are developed at AWI and
represent the testbed for the own approach in this paper. Representative input data for the experiments were provided by AWI. KH wrote the abstract and all sections of the paper.

**Competing interests**

The author declares that he has no conflict of interest.

**Acknowledgements**

I am grateful to Lars Ackermann (AWI) for information on the FESOM2-Iceberg development and Dr. Ying Ye (AWI), Dr. Martin Butzin (AWI), and Prof. Dr. Christoph Völker (AWI) for information on the FESOM2-REcoM development. Dr. Hendryk Bockelmann (DKRZ) has my thanks for providing the necessary context for performing the research within the PalMod project. I would also like to thank him for reviewing the paper and providing valuable suggestions and helpful comments.



**Financial support**

This research has been supported by the Federal Ministry of Education and Research/Bundesministerium für Bildung und Forschung (BMBF) for the PalMod project (Grant No.: 01LP1925A). This work used resources of the Deutsches Klimarechenzentrum (DKRZ) granted by its Scientific Steering Committee (WLA) under project ID bk0993.

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
