# Peer review of "Multiple execution of the same MPI application to exploit parallelism at hotspots with minimal code changes: a case study with FESOM2-Iceberg and FESOM2-REcoM"

_EGUsphere, 2023_

## Author Comment (AC1)

**Reply to RC2's comments on**

Multiple execution of the same MPI application to exploit parallelism at hotspots with minimal code changes: a case study with FESOM2-Iceberg and FESOM2-REcoM

RC2's comments are highlighted.

1)

RC2: My main concern is that I don't understand the underlying mechanism that allows for performance gains with multiexeMPI

Author answer:

Possibly there is too high an expectation of the complexity of MULTIEXECMPI. The complexity can be much higher with sophisticated domain decomposition implementation, for example. The new approach is indeed as simple as it is presented, especially in

line 205: "Figure 2. Exploiting parallelism at hotspot."

Figure 2 also contains the essential pseudo code.

Only a few changes in the manuscript will be necessary to make the basic idea of MULTIEXECMPI via improved pseudo code even clearer and to emphasize the strong analogy of the basic idea of MULTIEXECMPI to the parallelization of a loop based on OpenMP even more. In particular, the performance gains with MULTIEXECMPI are also achieved in an analogous way to a typical parallelization based on OpenMP, i.e. via splitting of the index range of a loop, but using processes with MultiExecMPI and threads with OpenMP for processing the subranges.

2)

RC2: It seems that the communication are enormous (the whole data must be synced) compared to traditional domain decomposition.

Author answer:

If "whole data" refers here to the entire simulation application, then it is exactly the redundant computations outside the parallelized loop that are consciously accepted with MULTIEXECMPI to prevent that the whole data must be synced.

For the explanation of MULTIEXECMPI using the example of the parallelization of the tracer loop of FESOM2-REcoM it is stated in

line 218: "To determine the data structures relevant for tracer data, simply those were picked out in the FESOM2-REcoM source code that are potentially write-accessed in the loop over all tracers

and are reused outside the loop. Basically, this is very similar to the procedure for a parallelization of the loop with OpenMP, where these data structures would have been declared as shared."

This also describes the data that must be synced (in the manuscript the term "merge" is used at the appropriate places). Of course, depending on a concrete simulation, even this subset of the total global data of a simulation application can still be a lot of data. In the manuscript there is the following consideration:

line 284: "For parallelizing a loop with MULTIEXECMPI there must be an appropriate relation between the computational and the communication effort, which is the more favorable the more can be computed in one iteration and the less data must be exchanged for the merge operations between the groups [to sync data]."

The comparison of the synchronization effort to traditional domain decomposition you mention in your comment is not meaningfully possible, because MULTIEXECMPI is not proposed as an alternative to a domain decomposition, but as a supplement, which is described at the appropriate places in the manuscript. Already in the abstract it is stated:

line 12: "Splitting the work at such hotspots between the instances represents an independent level of parallelization on top of the domain decomposition."

and

line 22: "Nevertheless, the implementation of the approach for other simulation models with similar properties seems promising, if the further reduction of the time-to-solution is in the focus, but a limit for the scalability based on the domain decomposition is reached."

As long as the application based on a domain decomposition approach still scales well (i.e. better than MULTIEXECMPI) when adding further computing power, the combination with MULTIEXECMPI does not yet make sense. In an analogous way, the use of OpenMP in the widespread hybrid approaches (i.e. using MPI processes for the domain decomposition and OpenMP threads for additionally parallelizing appropriate loops) would not make sense as long as the application based on a domain decomposition approach still scales well when adding further computing power.

The speedups of the experimental results show that MULTIEXECMPI works in practice and is quite equivalent to what one would expect from a parallelization based on OpenMP, e.g.:

line 504: "For the use of 2 FESOM2-REcoM groups, the speedup of 1.90, which results with respect to the tracer loop, can be estimated as quite good."

3)

RC2: Further, considering multiple groups and a decent strong scalability, the process in the groups could be employed to further fasten the computations with domain decomposition, instead they perform redundant computations. With a strong scalability of 0.75 and with 2 groups, the authors could get a speedup of 1.5 = (2*0.75) would they use the redundant group to actively participate in the comptuations.

Author answer

The same reasoning as under 2) applies here: The use of MULTIEXECMPI does not make sense as long as the application based on a domain decomposition approach still scales well (i.e. better than MULTIEXECMPI).

In the manuscript it is described as follows:

line 91: "With an increasing number of available cores to reduce the time-to-solution, the new approach can be used as a further level of parallelization, especially if the regions of a climate model can no longer be meaningfully broken down into smaller parts by means of the existing domain decomposition."

and

line 516: "With the steadily increasing number of cores per cluster node it can become increasingly difficult to utilize them merely on the basis of a domain decomposition. If a region can no longer be (meaningfully) broken down into smaller parts, the point is reached at which a combination with MULTIEXECMPI can utilize additional cores and achieve a further speedup."

If you are suggesting to extend the MultiExecMPI implementation to use the cores of group 2 (outside the parallelized loop) simultaneously for a domain decomposition, then this would not be possible because the purpose of the redundant computations is to prevent that the whole data must be synced – see also answer to 2). There would be no free computing resources available for this in MULTIEXECMPI. In addition, it would be a more complex extension anyway, which does not fit the character of MULTIEXECMPI to be a general method, whose attractiveness lies in the fact that only minimal code changes are required for implementation:

line 50: "In this paper a different approach is presented that provides an independent level of parallelism based on pure MPI communication and for which an existing simulation model can largely be considered as a black box."

4)

RC2: Please explain with a code example and resulting comunication size (on a dummy example) how it does help your comptations.

Author answer

A (dummy) code example for MULTIEXECMPI based on Figure 2 (line 205 in the manuscript) is given below (Fortran MPI). A short note first: because the MPI ranks start with index 0, my_group==0 corresponds to an MPI process in group 1, my_group==1 corresponds to an MPI process in group 2, and my_group==2 corresponds to an MPI process in group 3 in the manuscript:

```fortran
subroutine compute_tracers(mesh)
! compute_tracers is called in each simulation step
! i.e. loop over tracers below with num_tracers=33 using num_groups=3 is parallelized in each simulation step

call calc_slice(num_tracers, num_groups, my_group, tr_num_start, tr_num_end) ! split index range of 33 tracers
  ! Result of calc_slice is:
  ! tr_num_start=1,  tr_num_end=11 for an MPI process if it belongs to my_group==0 in its sub-domain
  ! tr_num_start=12, tr_num_end=22 for an MPI process if it belongs to my_group==1 in its sub-domain
  ! tr_num_start=23, tr_num_end=33 for an MPI process if it belongs to my_group==2 in its sub-domain

iteration_counter = 0
do tr_num = tr_num_start, tr_num_end

    call compute_tracer_data(tr_arr, tr_num) ! compute tracer data for current index tr_num
      ! Result of tracer computation is collected in a two-dimensional global array tr_arr at index position tr_num
      ! in the initialization phase of the application tr_arr was allocated via allocate(tr_arr(size_dim1, num_tracers))

    do group_i = 0, num_groups - 1 ! perform num_groups broadcast operations to merge (sync) tr_arr at the end of each tracer loop iteration

        call calc_slice(num_tracers, num_groups, group_i, tr_num_start_for_group_i, tr_num_end_for_group_i_dummy) ! calc start index of current group_i

        tr_num_to_send = tr_num_start_for_group_i + iteration_counter ! determine which part of tr_arr to broadcast using current iteration_counter as offset

        call MPI_Bcast(tr_arr(:, tr_num_to_send), size_dim1, MPI_DOUBLE_PRECISION, group_i, MPI_COMM_FESOM_SAME_RANK_IN_GROUPS, MPIerr)
          ! see line 185: "Figure 1. Multiple execution of an MPI application during startup phase" for implementation of MPI_COMM_FESOM_SAME_RANK_IN_GROUPS

    end do

    iteration_counter = iteration_counter + 1
end do

end subroutine compute_tracers
```

The communication pattern becomes immediately obvious with the code example. There are num_groups broadcast operations required per tracer loop iteration. The amount of data to be transmitted is determined by the size of tr_arr. The communication effort will have the more weight the larger size_dim1 is for a given value num_groups.

In practice, it will often be easy to find out whether the use of MULTIEXECMPI seems promising for an existing simulation application:

line 565: "Performance engineering techniques such as benchmarking and profiling and an analysis of the source code for data dependencies of compute-intensive loops are sufficient for this. For MPI calls, which may be contained within the loops, no additional (synchronization) effort is required anyway."

5)

RC2: Also, the first sections of the paper need a major refactoring. The author should focus on the actual contribution of the paper.

For example it's not clear to me the contribution of the multi-threaded MPI description, which also seems outdated.

Author answer

The manuscript reflects that first an OpenMP based parallelization of the tracer loop based on multi-threaded MPI was implemented for FESOM2-REcoM and only then MULTIEXECMPI was implemented as an alternative solution for parallelizing the tracer loop. In this way, the manuscript also vividly shows that in practice about the same speedups can be achieved with MULTIEXECMPI and OpenMP:

line 440: "Table 4. FESOM2-REcoM Speedups – MPI based groups vs. OpenMP threads."

A multi-threaded MPI solution is not outdated per se, but:

line 539: "An implementation based on MULTIEXECMPI in turn offers the advantage over the hybrid approach of avoiding dependencies from certain OpenMP features (e.g. a performant support of MPI_THREAD_MULTIPLE) and additional complexity with respect to the use of the compiler, MPI and runtime environment. In addition, it can be assumed for MULTIEXECMPI that debugging is also less complex than for a hybrid application (even if this is very well supported with powerful debuggers like arm DDT (Distributed Debugging Tool) (Arm, 2022))."

On the other hand, these descriptions take up some space. I am willing to follow your suggestion to shorten this part in the sense of a refactoring to improve the focus of the paper.